# Physician influence on medication adherence, evidence from a population-based cohort

**Shenzhen Yao[1][¤], Lisa M. Lix[2][☯], Gary Teare[3][☯], Charity Evans[1][☯], David F. Blackburn**[1]*

**1** College of Pharmacy and Nutrition, University of Saskatchewan, Saskatoon, Saskatchewan, Canada,
**2** Department of Community Health Sciences, Rady Faculty of Health Sciences, University of Manitoba,
Winnipeg, Manitoba, Canada, **3** Program Knowledge, Evidence and Innovation, Provincial Population and
Public Health, Alberta Health Services, Calgary, Alberta, Canada

☯ These authors contributed equally to this work.
¤ Current address: Vancouver Coastal Health Authority, Vancouver, British Columbia, Canada
* d.blackburn@usask.ca

University Faculty of Pharmaceutical Sciences,
THAILAND

**Data Availability Statement:** Our study was based
on health administrative data maintained by the
Government of Saskatchewan (Saskatchewan,
Canada). Access to these data was obtained

## Abstract

### Background

The overall impact of physician prescribers on population-level adherence rates are
unknown. We aimed to quantify the influence of general practitioner (GP) physician pre-
scribers on the outcome of optimal statin medication adherence.

### Methods

We conducted a retrospective cohort study using health administrative databases from Sas-
katchewan, Canada. Participants included physician prescribers and their patients begin-
ning a new statin medication between January 1, 2012 and December 31, 2017. We
grouped prescribers based on the prevalence of optimal adherence (i.e., proportion of days
covered $\geq$ 80%) within their patient group. Also, we constructed multivariable logistic
regression analyses on optimal statin adherence using two-level non-linear mixed-effects
models containing patient and prescriber-level characteristics. An intraclass correlation
coefficient was used to estimate the physician effect.

### Results

We identified 1,562 GPs prescribing to 51,874 new statin users. The median percentage of
optimal statin adherence across GPs was 52.4% (inter-quartile range: 35.7% to 65.5%). GP
prescribers with the highest patient adherence (versus the lowest) had patients who were
older (median age 61.0 vs 55.0, p<0.0001) and sicker (prior hospitalization 39.4% vs 16.4%,
p<0.001). After accounting for patient-level factors, only 6.4% of the observed variance in
optimal adherence between patients could be attributed to GP prescribers (p<0.001). The
majority of GP prescriber influence (5.2% out of 6.4%) was attributed to the variance unex-
plained by patient and prescriber variables.

through the Health Quality Council (HQC) in Saskatoon, Saskatchewan (www.saskhealthquality.ca). HQC's access to Saskatchewan databases is governed by a data sharing agreement with the province. Strict policies regulating access and reporting of these data are in place to protect the privacy of information for Saskatchewan citizens; thus, we are unable to make the data publically available. Inquiries regarding data access or verification can be directed to Dr. Tanya Verral, Program Director, Health Quality Council (tverrall@hqc.sk.ca).

**Funding:** The authors received no specific funding for this work.

**Competing interests:** David Blackburn is the Chair in Patient Adherence to Drug Therapy within the College of Pharmacy and Nutrition, University of Saskatchewan. This position was created through unrestricted financial support from AstraZeneca Canada, Merck Canada, Pfizer Canada, and the Province of Saskatchewan's Ministry of Health. None of the sponsors were involved in developing this study or writing the manuscript. This does not alter our adherence to PLOS ONE policies on sharing data and materials.

## Interpretation

The overall impact of GP prescribers on statin adherence appears to be very limited. Even "high-performing" physicians face significant levels of sub-optimal adherence among their patients.

## Introduction

Countless epidemiologic studies have constructed models to characterize poor medication adherence for a wide range of populations, disease groups, and specific drugs [1, 2]. Although countless associations between specific variables and medication adherence have been identified, few multivariable models account for more than a fraction of the overall variance in poor adherence observed at the population level [3]. One possible explanation for the poor performance of population adherence models is the frequent exclusion of physician prescribers. Physician prescribers influence adherence through several pathways including: [4–15] diagnosis [16], assessment [16, 17], prescribing [18–20], and providing education and follow-up [21–24]. In each of these roles, physicians may have opportunity to influence the knowledge, attitudes, tolerability, cost, and logistical barriers experienced by patients in starting a new drug [25, 26]. Thus, including the effect of prescribing physicians in population-based models may help account for the variance in adherence outcomes between patients.

Studies examining physician's impact on poor medication adherence have reported strong associations with prescriber communication, trust, frequency of visits, or organization of care [4, 5, 7–9, 11–15]. However, study design limitations often prevent attribution of these individual physician skills to adherence outcomes. For example, independent patient assessments of their physician's performance cannot be generalized to the experiences of other patients receiving that same physician's care. Nevertheless, if certain prescribing physicians communicate more effectively or establish trustful relationships more frequently than others, they would be expected to produce fewer cases of poor adherence via their positive effect on patient attitudes and beliefs. The influence of positive patient beliefs on medication adherence has been clearly established [27–30].

Measuring specific physician characteristics responsible for preventing poor adherence is challenging. Some physician characteristics change over time (e.g., age, experience, workload) while others are virtually impossible to define (e.g., intuition and interpersonal skills). However, evidence for the aggregate effect of unmeasured factors can be detected and quantified as latent effects using conventional modelling techniques [31].

The aim of this study was to quantify the extent to which individual general practitioner physician prescribers impact medication adherence in a population-based sample of new users of statin medications. In this study, general practitioner (GP) prescribers of statin medications were assessed because they prescribe over 85% of statin used in the provincial population [32] and are geographically distributed across urban and rural areas. Statin medications served as a robust model for medication adherence due to their simple dosage requirements, chronic indication, evidence-based benefits, and widespread use in adults.

## Methods

### Data sources

The study was conducted using administrative databases from Saskatchewan, Canada, which has a population of approximately 1.1 million [33] and a universal health care system. These

databases, linked by a common encrypted identification number for each patient, include the provincial health insurance registry file, the physician service claims file, the physician registry file, the hospital discharge abstract database, and prescription drug dispensation files [34]. The variables and data definitions of these files have been described in other studies [32, 35].

## Study design and population

We performed a retrospective cohort study of individuals receiving a first statin medication. Cohort members received at least one statin claim between January 1, 2012 and December 31, 2017 with no statin claims in the five years preceding the date of the earliest record (i.e., the index date). Cohort members were at least 18 years old on the index date and were continuously enrolled as beneficiaries of the provincial drug plan for at least five years before and one year following the index date. Patients were excluded from the cohort if: no GP prescribers were listed on their statin claims; missing values were observed for key variables (age/sex of patient or prescriber, country of medical training/graduation of prescriber, remuneration type for prescriber) [32]; hospitalized in an out-of-province acute care facility during the follow-up period; pregnancy within one year before or after the index date [International classification of diseases codes (ICD) 9th version (ICD-9): 641–676, V27; 10th version (ICD-10) and 10th revision of Canada (ICD-10-CA): O1, O21-95, O98, O99, Z37] or insufficient patient follow-up (i.e., termination of beneficiary status, death, long term care facility admission or reaching the study end date of December 31, 2018 in the 5 years before, or one year after the index date).

For each statin user, a single GP prescriber was identified by selecting the physician listed on highest number of statin claims during the one-year follow-up period (after excluding claims listed with specialist prescribers). In rare situations where two prescribers were listed on the same number of statin claims, one was selected at random.

## Outcome measure

The study outcome was optimal adherence to statin medications during the first year of therapy, defined by the proportion of days covered (PDC) $\geq$80% [25, 26]. The PDC was estimated using the sum of the number of tablets dispensed (assuming one tablet per day dosing) divided by the number of days in the follow up period (365 days) adjusted for any days spent in a hospital (i.e., because drug dispensations are not captured for hospitalized patients) [36]. Tablets dispensed during early refills were allowed to accumulate in the numerator and switching between different statin medications was allowed.

## Explanatory variables

The variables of interest included fixed and time-varying characteristics of GP prescribers. Fixed prescriber level variables included sex and country of medical training. Time-varying prescriber level variables included prescriber's age, years in practice, remuneration type [i.e., fee-for-service (FFS) or non-fee-for-service (Non-FFS)], overall patient count (as a proxy for prescriber workload) [37, 38]. and statin patient count to indicate a GP's experience with statin medications [Appendices A and B in S1 File].

We also included numerous patient-level variables previously used in medication adherence studies [2] to minimize confounding: age, sex, rural/urban residence [39]; calendar year of index date; neighborhood median household income quintiles [40, 41]; clinical and health services records in the year prior to index (i.e., number of distinct prescription medication classes received [42], number of outpatient service claims, percentage of prescription medication cost paid by government health insurance, number of hospitalizations for acute care, number of emergency department visits, Charlson comorbidity score [43], and presence or

absence of numerous clinical conditions using validated case definitions by the Canadian Chronic Disease Surveillance System [Appendix B in S1 File] [44]. We also included an indicator of continuity of care, which was strongly associated with medication adherence in a previous study using the present cohort [35].

## Statistical analysis

For each GP prescriber, we calculated the prevalence of optimal statin adherence within their patient group and ranked prescribers into quartiles of increasing prevalence of statin adherence. We described patient and GP physician prescriber characteristics of the overall cohort and within each quartile. Between-group differences for median values were assessed by the Wilcoxon rank-sum test, and percentages by the Chi-squared test.

Next, we quantified the influence of GP prescribers (independent of patient characteristics) with multivariable logistic regression analyses using two-level (patient and prescriber) non-linear mixed-effects models. We calculated the intraclass correlation coefficient (ICC) for GP physician prescribers from two models. Model A (the overall physician effect) included prescriber identification numbers (i.e., random intercept) plus patient-level variables (detailed below). Model B (the latent prescriber effect independent of prescriber characteristics) included prescriber-level characteristics, the random intercept term, and patient variables [31]. All patient-level variables were entered in these models except for those exhibiting multi-collinearity with any of the prescriber-level factors (i.e., variance inflation factor > 2.5) [45].

Each prescriber-level variable was added individually and assessed for improvements in goodness of fit statistics using the likelihood ratio test (LRT) [46]. For time-varying prescriber-level variables, we evaluated multiple possible components including a contextual effect (between prescribers), a compositional effect (between patients within a prescriber), a random slope (the compositional effect varying between prescribers), and between/within level interactions [46]. The mean centering method was used to decompose these effect components [46–48]. Prescriber age and calendar year on index date was excluded from all models due to a strong correlation with years in practice with no evidence of interaction. SAS statistical software, version 9.4, (SAS Institute Inc., Cary, NC, USA) was used to conduct all analyses [49].

## Ethical considerations

Ethics approval was granted by the University of Saskatchewan Biomedical Research Ethics Board (14–143). Data was accessed at the Saskatchewan Health Quality Council under data sharing agreements with the Saskatchewan Ministry of Health and eHealth Saskatchewan.

## Results

We identified 58,549 patients who initiated statin therapy between January 1, 2012, and December 31, 2017. Among them, 3,405 (5.8%) were excluded for residing in a long-term care setting, hospitalized in an out-of-province facility, pregnancy diagnosis, or having no service claims by a GP during the follow-up period. Also, 3,270 (5.5%) patients were excluded because their statin prescriber was missing data on birth, sex, graduation, country of medical training, or remuneration type. The final cohort included 51,874 new users paired to 1,562 GP prescribers [Fig 1].

The median age of patients on the index date was 59.0 years (IQR 51.0/67.0), 43.9% were female, 30.5% (n = 15,830) lived in a rural area, 32.7% (n = 16,988) had a Charlson score greater than zero, and 22.2% (n = 11,493) received acute care in hospital within 365 days prior to the index date. The median age of prescribers on the index date was 50.0 years (IQR 40.0/49.0), 26.1% (n = 13,532) of patient-prescriber pairs included a female prescriber, and 29.8%

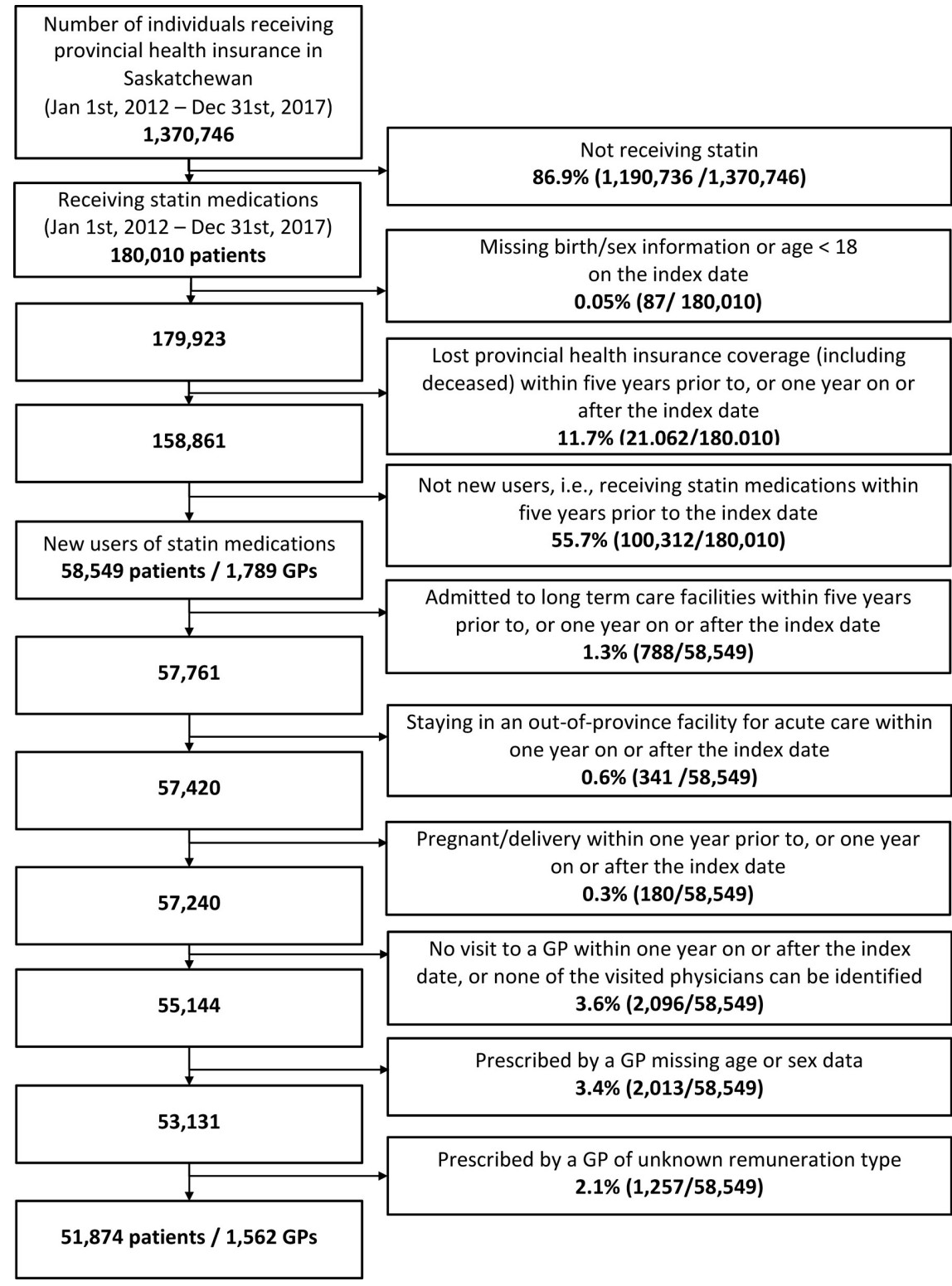

**Fig 1. Study flow chart.**

(n = 15,462) included a prescriber trained in Canada. Prescribers cared for a median of 3,346 (IQR 2,203/5,453) patients and 276 (IQR 177/413) patients who used statins. The median number of new statin users under each prescriber was 16 (upper quartile boundary 43 but lower boundary suppressed due to cell size < 6).

The median prevalence of optimal statin adherence for patients grouped under individual GP prescribers was 52.4% (IQR 35.7%/65.4%). After ranking prescribers into quartiles based on increasing prevalence of statin adherence, clear differences in patient characteristics were observed between prescriber groups [Table 1]. Compared to GP prescribers in the lowest quartile of patient adherence, those with the highest prevalence of optimal adherence (i.e., highest quartile) had patients who were older (median age in years = 61.0 IQR 54.0/70.0 vs 55.0 IQR 47.0/64.0, p<0.001), less likely to be female (39.9% vs 47.4%, p<0.001), more likely to have a previous hospitalization for acute care (39.4% vs 16.4%, p<0.001) or emergency room visit (30.1% vs 19.9%, p<0.001), more visits to a specialist (median 4.0 (IQR 1.0/10.0) vs 2.0 (IQR 0.0/5.0), p<0.0001), and more with a Charlson score greater than zero (46.4% vs 26.9%, p<0.001). Prescriber characteristics also differed across these quartiles. Prescribers in the highest quartile were less likely to be female (21.9% vs 27.9%, p<0.001), more likely to be trained in Canada (55.7% vs 12.2%, p<0.001), more likely to be a FFS prescriber (17.9% vs 14.0%, p<0.001), and provided care to more statin users (median statin patient count = 253 IQR 176/ 334 vs 210 IQR 116/350, p<0.001) [Table 1].

A GP prescriber's country of medical training was significantly associated with their patient's odds of optimal statin adherence (Canada vs foreign, uOR = 1.53, 95%CI 1.47 to 1.59; aOR = 1.40, 95%CI 1.30 to 1.51). However, compared to patients receiving statins from Canadian-trained GPs (n = 15,452), those prescribed statins by foreign trained GPs (n = 36,422) were more frequently living in rural areas (32.4% vs 26.1%, p<0.001), had a substantially lower incidence of prior hospitalizations (18.1% vs 31.8%, p<0.001), fewer emergency department visits (17.5% vs 29.7%, p<0.001), and fewer with a Charlson score greater than 0 (i.e., score >1 = 29.0% vs 41.7%, p<0.001) [Appendix C in S1 File].

A similar finding was observed for prescribers categorized as non-FFS (versus FFS) remuneration type. In the adjusted analysis, non-FFS was significantly associated with an increased odds of optimal adherence (aOR = 1.18, 95%CI 1.08 to 1.29); however, the unadjusted estimate suggested the opposite effect (uOR = 0.94, 95%CI 0.90 to 0.99). Again, this variable appeared highly confounded when examining patient characteristics as well as the distribution of patients across these two types of prescribers (7,849 statin patients prescribed by non-FFS GPs vs 44,025 by FFS GPs). Patients prescribed statins by non-FFS GPs were more often living in a rural area (47.6% vs 27.5%, p<0.001), less likely to have been hospitalized previously (17.5% vs 23.0%, p<0.001) or having visited an emergency department (14.8% vs 22.2%, p<0.001), and fewer having a Charlson score greater than zero (26.4% vs 33.9%%, p<0.001). Further, non-FFS prescribers had fewer years in practice (median years 15.0, IQR 9.0/27.0 vs 25.0, IQR 15.0/ 34.0%, p<0.001), and lower overall patient counts (median = 2,112, IQR 1,567/ 2,800 vs 3,720, IQR 2409/5,823, p<0.001).

The overall effect of the GP prescriber's years in practice was not significantly associated with optimal adherence (i.e., per ten additional years in practice: uOR = 0.98, 95%CI 0.96 to 0.99; aOR = 0.98, 95%CI 0.96 to 1.01). However, decomposition of the overall effect into the between-prescriber effect (i.e., per ten additional years in practice: aOR = 0.76, 95% CI 0.66 to 0.87) and the within-prescriber effect (per ten additional years in practice: aOR = 1.30, 95%CI 1.14 to 1.48) appeared to be contradictory. Additional analyses using dispersion to represent the within-prescriber effect confirmed the existence of these two effects simultaneously [Appendices D and E in S1 File].

**Table 1. Patient and prescriber characteristics.**

| Characteristic | Total (all patients) | Prescriber Quartile 1 | Prescriber Quartile 2 | Prescriber Quartile 3 | Prescriber Quartile 4 |
|---|---|---|---|---|---|
| | | *(% of patients with optimal statin adherence <35.7%)* | *(% of patients with optimal statin adherence 35.7% to 52.2%)* | *(% of patients with optimal statin adherence 52.3% to 65.4%)* | *(% of patients with optimal statin adherence >65.4%)* |
| Prescribers (n) | n = 1,562 | n = 393 | n = 387 | n = 391 | n = 391 |
| Patients(n) | n = 51,874 | n = 6,251 | n = 14,640 | n = 19,760 | n = 11,223 |
| % of patients with optimal adherence[a] | 53.6 | 25.5 | 45.3 | 57.8 | 72.5 |
| **Patient characteristics** | | | | | |
| Age, median (IQR[b]) | 59.0 (51.0, 67.0) | 55.0 (47.0, 64.0) | 58.0 (50.0, 66.0) | 59.0 (52.0, 68.0) | 61.0 (54.0, 70.0) |
| Females, n(%) | 22,781 (43.9) | 2,966 (47.4) | 6,669 (45.6) | 8,669 (43.9) | 4,477 (39.9) |
| 1+ hospitalizations for acute care, n(%) | 11,493 (22.2) | 1,025 (16.4) | 2,284 (15.6) | 3,765 (19.1) | 4,419 (39.4) |
| Visits to GPs[c], median (IQR) | 6.0 (3.0, 9.0) | 6.0 (3.0, 10.0) | 5.0 (3.0, 9.0) | 6.0 (3.0, 9.0) | 5.0 (3.0, 9.0) |
| Visits to specialists, median (IQR) | 2.0 (0.0, 6.0) | 2.0 (0.0, 5.0) | 2.0 (0.0, 5.0) | 2.0 (0.0, 6.0) | 4.0 (1.0, 10.0) |
| 1+ visits to emergency department, n(%) | 10,952 (21.1) | 1,243 (19.9) | 2,531 (17.3) | 3,802 (19.2) | 3,376 (30.1) |
| Income level, n(%) | | | | | |
| 1 | 9,569 (18.4) | 1,608 (25.7) | 2,826 (19.3) | 3,275 (16.6) | 1,860 (16.6) |
| 2 | 9,500 (18.3) | 1,224 (19.6) | 2,813 (19.2) | 3,599 (18.2) | 1,864 (16.6) |
| 3 | 9,540 (18.4) | 1,046 (16.7) | 2,675 (18.3) | 3,645 (18.4) | 2,174 (19.4) |
| 4 | 10,685 (20.6) | 1,167 (18.7) | 2,920 (19.9) | 4,222 (21.4) | 2,376 (21.2) |
| 5 | 9,782 (18.9) | 848 (13.6) | 2,603 (17.8) | 4,032 (20.4) | 2,299 (20.5) |
| missing | 2,798 (5.4) | 358 (5.7) | 803 (5.5) | 987 (5.0) | 650 (5.8) |
| Rural residence, n(%) | 15,830 (30.5) | 1,923 (30.8) | 4,568 (31.2) | 6,173 (31.2) | 3,166 (28.2) |
| Charlson score > 0, n(%) | 16,988 (32.7) | 1,683 (26.9) | 3,946 (27.0) | 6,148 (31.1) | 5,211 (46.4) |
| **Prescriber characteristics[f]** | | | | | |
| Caseload[d], median (IQR) | 16 (<6[g], 43) | 6 (<6, 14) | 21 (7, 49) | 33 (17, 67) | 10 (<6, 35) |
| Age, median (IQR) | 50.0 (40.0, 49.0) | 55.0 (43.0, 67.0) | 48.0 (40.0, 59.0) | 49.0 (38.0, 58.0) | 50.0 (41.0, 57.0) |
| Female, n(%) | 13,532 (26.1) | 1,746 (27.9) | 4,205 (28.7) | 5,125 (25.9) | 2,456 (21.9) |
| Medical training in Canada, n(%) | 15,462 (29.8) | 765 (12.2) | 2,405 (16.4) | 6,038 (30.6) | 6,254 (55.7) |
| Non-FFS[e] remuneration, n(%) | 7,849 (15.1) | 1,122 (17.9) | 2,510 (17.1) | 2,648 (13.4) | 1,569 (14.0) |
| Years in practice, median (IQR) | 24.0 (13.0, 33.0) | 28.0 (16.0, 42.0) | 22.0 (13.0, 32.0) | 24.0 (12.0, 32.0) | 25.0 (15.0, 32.0) |
| Overall patient count, median (IQR) | 3,346 (2,203, 5,453) | 3,535 (2,080, 5,745) | 3,313 (2,217, 5,491.5) | 3,273.5 (2,249.5, 5,362.5) | 3,390 (2,120, 5,474) |
| Statin patient count, median (IQR) | 276 (177, 413) | 210 (116, 350) | 289 (170, 418) | 309 (206, 458) | 253 (176, 334) |

[a]Optimal adherence = proportion of days covered > = 80% of statin medications

[b]IQR = interquartile range

[c]GP = general practitioners

[d]Caseload = number of study patient (new statin users) per prescriber

[e]Non-FFS = non-fee-for-service remuneration type

[f]index date = patient's first statin dispensation date

[g]<6: actual number of patients was suppressed as there were less than six patients in the group. Patient and physician characteristics measured within 365 days prior to the date of the first dispensation of a statin (index date), or on the index date, except that overall patient count, and statin patient count were measured within 365 days prior to and 365 days on and after the index date.

**Table 2. Odds ratios (95% confidence interval) for the association of prescriber-related characteristics with optimal statin adherence (proportion of days covered by statin medications > = 80%).**

| Prescriber characteristics[a] | Unadjusted odds ratio (95%CI[c]) | Adjusted[d] odds ratio (95%CI) | | |
| --- | --- | --- | --- | --- |
| | | Between prescribers | Within a prescriber | Random slope |
| Country of medical training (Canada vs foreign) | **1.53 (1.47, 1.59)[e]** | **1.40 (1.30, 1.51)** | | |
| Sex (female vs male) | **0.93 (0.90, 0.97)** | 0.99 (0.91, 1.07) | | |
| Years in practice (per 10 years increase) | **0.98 (0.97, 0.99)** | **0.76 (0.66, 0.87)** | **1.30 (1.14, 1.48)** | **1.30 (1.26, 1.35)** |
| Remuneration type (Non-FFS vs FFS)[b] | **0.94 (0.90, 0.99)** | **1.18 (1.08, 1.29)** | 1.23 (0.91, 1.66) | |
| Overall patient count (per 1,000 increase) | **0.98 (0.98, 0.99)** | **0.98 (0.97, 1.00)** | 0.99 (0.96, 1.01) | |
| Statin patient count (per 100 increase) | 1.01 (1.00, 1.02) | **1.06 (1.03, 1.09)** | 1.02 (0.97, 1.06) | |

[a]Country of medical training and sex were measured on the date of the first dispensation of a statin (index date), overall patient count and statin patient count measured on 365 days prior and 365 days on and after the index date

[b]Non-FFS = non-fee-for-service remuneration method, FFS = fee-for-service remuneration method

[c]95%CI = 95% confidence interval

[d]Adjusted for patient variables including age, sex, urban/rural living, household income level, number of medications by the anatomical therapeutic chemical (ATC) class, number of outpatient visits, percentage of medication cost paid by government health insurance, number of hospitalization for acute care, number of visits to emergency department, Charlson comorbidity score, clinical conditions (osteoporosis, rheumatoid arthritis, hypertension, stroke, ischemic heart disease, acute myocardial infarction, heart failure, multiple sclerosis, Parkinson's disease, Alzheimer's disease and dementia, epilepsy, asthma, chronic obstructive pulmonary disease, diabetes, mood and anxiety diseases, schizophrenia, and cancer); also adjusted for prescriber-related variables in the table

[e]Odds ratios in bold font are statistically significant (p<0.05).

Finally, a small but positive association with optimal adherence was observed for GP prescribers with higher number of statin patients (i.e., for every additional 100 statin patients uOR = 1.01, 95%CI 1.00 to 1.02; aOR = 1.06, 95%CI 1.03 to 1.09). In contrast, total patient count of a prescriber (i.e., representing workload) showed a very slight negative association with the odds of achieving optimal adherence (i.e., for 1000 additional patients uOR = 0.98, 95%CI 0.98 to 0.99; aOR = 0.98, 95%CI 0.97 to 1.00). Prescriber sex was not significantly associated with patients' adherence outcomes (uOR = 0.93, 95%CI 0.90 to 0.97; aOR = 0.99, 95%CI 0.91 to 1.07) [Table 2].

Based on the ICC from the model representing the overall prescriber effect (i.e., including patient level variables and the random effect), individual prescribers accounted for 6.4% of the total variance in optimal statin adherence observed in the population. After accounting for all patient and prescriber variables, the prescriber latent effect accounted for 5.2% of the variance in optimal adherence among the study population (reduced by 18.8% from the overall prescriber variance, p<0.001).

## Discussion

In this population-based study, we examined the impact of GP prescribers on patient adherence to statin medications. When prescribers were ranked into quartiles based on their patient's optimal adherence rate, several notable observations were evident. First, the upper quartile boundary of optimal adherence was only 65.4%. In other words, it appeared that "performance" on medication adherence was not skewed and many of the highest 'performing' prescribers still failed to support optimal adherence in up to one-third of their patients. Furthermore, prescribers with the highest prevalence of optimal adherence cared for patients who were older, sicker, and more likely to include a specialist in their care. Thus, after controlling for patient characteristics, differences in adherence associated with GP prescribers were virtually eliminated. Ultimately, GP prescribers only affected 6.4% of the optimal adherence outcomes observed in the study population. An extensive body of research suggests prescriber

characteristics such as superior communication skills is associated with higher patient adherence [4, 5, 7, 12–14]. However, our results suggest observed associations with physician skills such as communication may have been a result of reverse causality bias where an individual prescriber is likely to be rated highly by adherent patients only (i.e., also rated poorly by non-adherent patients). Surveys studies are typically anonymous so cannot identify this source of bias through specific provider linkage.

The influence of physician remuneration strategy on patient care, efficiency, and healthcare sustainability is an important issue for healthcare policy makers [50]. Theoretically, non-FFS physicians spend more time with patients resulting in increased patient satisfaction and quality of care [51]. In our study, patients receiving statins from non-FFS prescribers exhibited higher odds of optimal adherence. However, the number of Non-FFS practitioners were far lower than FFS in our population and the characteristics of patients were different also. Patients receiving care from non-FFS prescribers were highly skewed towards living in rural areas with lower levels of comorbidity. As a result, it appears the distribution of non-FFS prescribers in our province is not adequate to allow non-randomized evaluation without substantial uncertainty due to the role of bias. Further study is needed to quantify the benefits and weaknesses of non-FFS remuneration models on medication adherence. Quantitative evidence for the impact of remuneration models is lacking overall [50, 52, 53].

The impact of a prescriber's years in practice was complex but decomposition revealed important findings. The absolute number of years in practice had a significant, albeit relatively small negative impact on a prescriber's ability to influence optimal adherence. In a systematic review of 62 studies between 1966 and 2004 by Choudhry and colleagues, 32 (52%) studies reported that clinical knowledge, adherence to diagnosis and treatment guidelines, and patient health outcome declined as years in practice increased [54]. However, the within-prescriber analysis suggested that the density of statin prescribing throughout the years of practice was also an important factor. Specifically, GPs starting new statin prescriptions across multiple years tended to have more adherent patients than those initiating statin treatment less frequently. We also found that the absolute number of statin patients under a GP prescriber's care was associated with a higher odds of optimal adherence. These findings demonstrate the difference between the absolute impact of years in practice versus the influence of continued activity with statins during those years of practice. It is plausible that frequent prescribing of statin medications throughout the course of a prescriber's career would improve their skills and experience in positively influencing a patient's adherence behaviour.

Our study had limitations. First, we only captured dispensations but not consumption of statins. However, dispensation data have been widely used to estimate medication adherence with high validity [55]. Second, lack of randomization limited our control over unmeasured confounding. This limitation appeared to be especially problematic for assessments of remuneration type and country of medical training. Devlin and colleagues reported that physicians may self-select into a remuneration type due to uncaptured personal preference and characteristics [56]. Third, the databases used for our study lack information about clinical factors (e.g., cholesterol levels), lifestyle factors (e.g., smoking), and some adherence determinants (e.g., health literacy, patient knowledge). Thus, despite including a high number of potential confounders, we were unable to capture some possible confounders and may have only partially addressed others (e.g., specific comorbidities). Fourth, although the process for identifying the statin prescriber was successful in over 95% of patients, we were unable to identify those who switched prescribers during the follow-up period. Finally, the impact of calendar year was a possible confounder in our analysis as adherence to many chronic medications has been increasing for years [57]. However, it was excluded from the present analysis because it was

highly correlated to years in practice. Ultimately, the likelihood that unmeasured confounding concealed an important influence of prescribing physicians was low in our view.

## Conclusion

The overall impact of GP physician prescribers on their patient's adherence is very limited. Even "high-performing" prescribers face significant levels of sub-optimal adherence among their patients.

## Supporting information

**S1 File.**
(DOCX)

## Acknowledgments

David Blackburn is the Chair in Patient Adherence to Drug Therapy within the College of Pharmacy and Nutrition, University of Saskatchewan. This position was created through unrestricted financial support from AstraZeneca Canada, Merck Canada, Pfizer Canada, and the Province of Saskatchewan's Ministry of Health. None of the sponsors were involved in developing this study or writing the manuscript.

This study is based in part on de-identified data provided by the Saskatchewan Ministry of Health. The interpretation and conclusions contained herein do not necessarily represent those of the Government of Saskatchewan or the Saskatchewan Ministry of Health.

## Author Contributions

**Conceptualization:** Shenzhen Yao, David F. Blackburn.

**Formal analysis:** Shenzhen Yao, Lisa M. Lix.

**Investigation:** Shenzhen Yao.

**Methodology:** Shenzhen Yao, Lisa M. Lix, Gary Teare, Charity Evans, David F. Blackburn.

**Project administration:** David F. Blackburn.

**Supervision:** Lisa M. Lix, Gary Teare, Charity Evans, David F. Blackburn.

**Validation:** Shenzhen Yao, Gary Teare, Charity Evans, David F. Blackburn.

**Visualization:** Lisa M. Lix, Gary Teare, Charity Evans.

**Writing – original draft:** Shenzhen Yao.

**Writing – review & editing:** Shenzhen Yao, Lisa M. Lix, Gary Teare, Charity Evans, David F. Blackburn.

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
