## [Decision Letter · Decision Letter 0]

4 Oct 2022

PONE-D-22-22029Physician influence on medication adherence, evidence from a population-based cohort.PLOS ONE

Dear Dr. Blackburn,

Thank you for submitting your manuscript to PLOS ONE. After careful consideration, we feel that it has merit but does not fully meet PLOS ONE’s publication criteria as it currently stands. Therefore, we invite you to submit a revised version of the manuscript that addresses the points raised during the review process.

The paper in general is methodologically sound for its internal validity concern and exhibits merit to medication adherence field. To enhance clarity of the paper, the issues addressed by the reviewers  are advised to be revised.

We look forward to receiving your revised manuscript.

Kind regards,

Chulaporn Limwattananon, Ph.D.

Academic Editor

PLOS ONE

Journal Requirements:

2. Please ensure that you have specified (1) whether consent was informed and (2) what type you obtained (for instance, written or verbal, and if verbal, how it was documented and witnessed). If your study included minors, state whether you obtained consent from parents or guardians. If the need for consent was waived by the ethics committee, please include this information.

   "David Blackburn is the Chair in Patient Adherence to Drug Therapy within the College of Pharmacy and Nutrition, University of Saskatchewan.  This position was created through unrestricted financial support from AstraZeneca Canada, Merck Canada, Pfizer Canada, and the Province of Saskatchewan’s Ministry of Health.  None of the sponsors were involved in developing this study or writing the manuscript. "

Additional Editor Comments:

Methods

Page 6, paragraph 2 on study design and population:

How did a patient who had multiple statin prescribers over the one-year follow up period was handled?  Are there any covariates capturing the cases for multiple prescribers or multiple statin types, especially for the patients who experienced switching of the prescribers, statins, or both were included in the models?

Pages 6-7, outcome measures

Did the dataset contain dosage strength and regimens of the prescribed statins? If that’s so, why not the defined daily doses were used instead of number of tablets for estimating the PDCs.

Page 7, explanatory variables

Apart from demographic characteristics, socio-economic status, and clinical conditions of the study patients, “health literacy” is likely to play an important role on medication adherence. Are there any explanatory variables capturing the patient’s health literacy that were included in the model?  It would be good if the paper would distinguish the explanatory variables that classified patient and prescriber characteristics in a systematic way.

Results

Table 1.  It would be good if the adherence outcome would be presented in terms of mean +/- SD or median +/- IQR of PDCs in addition to % patients with optimal (>80% PDCs) adherence.

Reviewers' comments:

Reviewer's Responses to Questions

**Comments to the Author**

1. Is the manuscript technically sound, and do the data support the conclusions?

Reviewer #1: Partly

Reviewer #2: Yes

2. Has the statistical analysis been performed appropriately and rigorously? 

Reviewer #1: Yes

Reviewer #2: Yes

3. Have the authors made all data underlying the findings in their manuscript fully available?

Reviewer #1: No

Reviewer #2: Yes

4. Is the manuscript presented in an intelligible fashion and written in standard English?

Reviewer #1: Yes

Reviewer #2: Yes

5. Review Comments to the Author

Reviewer #1: 1) Title: in this study...physician means only GP physician then the title might be focused on GP, please consider.

2) Abstract: Please show the main results of the GP characteristics that influence on medication adherence and rewriting conclusion

3) Introduction: The Rationale for doing research should be strong enough even there is unknown, but it should show the value of the information of this research which is gap of the study such as policy initiative.

4) Methods: It needs more detail about 1) PDC meaning for sum of days covered in time frame, is it the same as days dispensed or sum of days supply 2) ORs both uOR and aOR and 95% CI of OR 3) you described Wilcoxon and Chi square for statistical testing, but there no evidence that show where to use these statistics, please show in the results or making a table for this part for understanding easily 4) you described about Model A and B, please write the results that related to models or show in the results about the models.

3) Results: I recommend making a table to show and summarize the main points of the statistical significance, might make the readers understand easily.

5) Discussion: Consider rewriting discussion: writing key findings of the study at the start of the Discussion and then discuss the important findings of the study in the Discussion and how it relates to literature. This can help researchers explain what this study adds to the overall body of literature. Limitations, as in your introduction "Physician prescribers influence adherence through several pathways including:4-15 diagnosis,16 assessment,16,17 prescribing,18-20 and

providing education and follow-up.21-24", please explain the connection to the explained variance of med adherence and your study. And give some recommendations for the future study or other researchers in this area.

6) Conclusion: consider rewriting conclusion that answers your question or objective.

Reviewer #2: Reviewer #1: Physician influence on medication adherence, evidence from a population-based cohort.

Authors report an interesting study about physician influence on medication adherence, evidence from a population-based cohort. There are many studies that modeled and verified patient self -management levels and regional characteristics of drug compliance, but there are few studies that analyze the behavioral effects of doctors.

Methods:

It is difficult to conclude because some data are missing such as patient disease duration, prior patient’s disease management education experiences and distribution of comorbidity. It is need to describe some limitation.

In order to prevent unnecessary complications, there were many arbitration and research that combines various behavior theories that allow patients to maintain healthy behavior (like medication compliance). Authors should report some publications.

Also, training counseling experience of physician for compliance can have a positive impact on the patient's behavioral changes. Author should report the indicator (eg. Years in practice in this paper) can use as a proxy indicator for educational skill to improve the self efficacy.

Results:

Some data are missing. (Table1)

- Distribution of different age periods among the older patients: <50 50-64, 65-75, >75 years old.

- Type of comorbidity or diagnostic among patients.

Table2 :

The study outcome was optimal adherence to statin medications during the first

year of therapy, defined by the proportion of days covered (PDC) ≥80%.

Author quantified the influence of GP prescribers for optimal adherence group(independent of patient characteristics) with multivariable logistic regression analyses. But there is no multivariate result for patient group with bad adherence and need to add some table.

6. PLOS authors have the option to publish the peer review history of their article (what does this mean?). If published, this will include your full peer review and any attached files.

Reviewer #1: No

Reviewer #2: No

---

## [Author Response · Author response to Decision Letter 0]

11 Oct 2022

Thank you for the review. We have uploaded itemized responses to each request by the reviewers. Please let us know if you need anything else.

---

## [Editor Report · Decision Letter 1]

17 Nov 2022

Physician influence on medication adherence, evidence from a population-based cohort.

PONE-D-22-22029R1

Dear Dr. Blackburn,

We’re pleased to inform you that your manuscript has been judged scientifically suitable for publication and will be formally accepted for publication once it meets all outstanding technical requirements.

Kind regards,

Chulaporn Limwattananon, Ph.D.

Academic Editor

PLOS ONE
---

## [Editor Report · Acceptance letter]

22 Nov 2022

PONE-D-22-22029R1 

Physician influence on medication adherence, evidence from a population-based cohort. 

Dear Dr. Blackburn:

I'm pleased to inform you that your manuscript has been deemed suitable for publication in PLOS ONE. Congratulations! Your manuscript is now with our production department. 

Kind regards, 

on behalf of

Dr. Chulaporn Limwattananon 

Academic Editor

PLOS ONE